# Wolf and Dog: What Differences Exist?

**Alessandra Coli** [1,*], **Davide Prinetto** [2] **and Elisabetta Giannessi** [1]

1   Department of Veterinary Sciences, University of Pisa, 56124 Pisa, Italy
2   Veterinary Anatomy Museum, University of Pisa, 56124 Pisa, Italy
*   Correspondence: alessandra.coli@unipi.it; Tel.: +39-050-221-3856

**Abstract:** A morphological study of the skeletal specimen of *Canis lupus* L. from an archeological dig of Agnano (Pisa) (Fauna Laboratory, Department of Archaeological Sciences, University of Pisa, Italy) that is chronologically placed in the Wurm period (last glaciation) was done to perform an anatomical comparison between this wild ancestor and osteological specimens of *Canis familiaris* L. present in the Veterinary Anatomy Museum (University of Pisa). Marked morphological differences in the splanchnocranium (nasal bone, zygomatic arch and orbital angle), neurocranium (sagittal crest) and temporomandibular joint (due to different developments of the masticatory muscles) are highlighted on the wolf specimen compared to those in the domestic dog specimens present in Museum. The appendicular skeletal bones of the wolf show anatomical features similar to those of dog bone specimens, confirming their belonging to the same family (*Canidae*). This result confirms that domestication has almost exclusively affected the anatomical features of the skull that have changed due to the difference in dietary approach between wolves and dogs.

**Keywords:** domestication; skeleton; morphological study; *Canidae*

## 1. Introduction

The wolf (*Canis lupus* L., 1758) belongs to the *Canidae* family, whose main morphological characteristics are a long dental row, a large number of teeth (42), a long tail, digitigrade limbs and four fingers in the hind limb. The wolf is considered to be the wild ancestor of the domestic dog (*Canis lupus familiaris* L., 1758). The anatomical similarity between wild and domesticated species is actually considered important in the study and monitoring of infectious diseases' spread [1]; for instance, the domestic dog is actually considered to be the main reservoir of the Canine Distemper Virus (CDV) [2], thanks to epidemiological monitoring using Geographical Information System (GIS) [3].

In Italy, Altobello [4] highlighted characteristics to distinguish the Italian wolf from the populations of other European wolves, regarding it a subspecies of the gray wolf (*Canis lupus* L., 1758) and calling it "Apennine wolf". Recent genetic investigations have confirmed this statement by elevating the Apennine wolf to subspecies (*Canis lupus italicus*), thus distinguishing it, by morphological and genetic characteristics, from the remaining populations of European wolves [5]. From a genetic point of view, the gray wolf represents the original dog line in Asia and Europe. A case of hybridization between wolf and dog was reported in Europe, and its occurrences were well analyzed by genetic analysis of the wolf–dog hybrids in several European countries [6]. The Apennine wolf represents a wolf colony that recolonized the western Italian Alps, as reported [7] by analyzing the DNA extracted from Apennine wolf tissue samples and genotyping at 12 microsatellite loci. From this genetic study, it can be stated that the Apennine wolf has a significantly higher heterozygosity than the those in the wolves from the Alps. Currently the wolf distribution affects the whole Apennine chain with branches in Lazio and Tuscany, which has a total population estimated at 400–500 individuals [8]. The explanation for where and when this animal domestication took place remains surprisingly inaccurate [9]. Scientific studies show that it was not man who sought the wolf, transforming it into a dog, but

the exact opposite; the wolf approached human settlements to eat the remains of meals, losing fear of man over time and making itself tamable [10]. It is therefore a process of "self-domestication", wherein wolves and men shared an ecological niche, allowing the wolf to become domesticable and showing changes in the behavioral and morphological features [11]. In fact, the bones can show the morphological and structural transformations due to the domestication process that occurred over 14,000 ears ago, for instance, in the bones of the primitive *Canis lupus familiaris* that were found in a Pleistocene archaeological excavation in a human burial. However, results based on the study of mitochondrial DNA have suggested that the domestication process can be traced back as early as 40,000 years ago [12].

From the literature, the study of the wolf skeleton was particularly focused on the morphological features of the skull; it reached a length of about 23–27 cm and a width of about 15–18 cm, while the dog skull has a different length and width in relation to the breed.

A long splanchnocranium, large zygomatic arches and a developed external sagittal crest are particularly developed in the wolf skull [13]. The angle formed by the intersection between the straight-line tangent to the top of the skull and the tangent line to the zygomatic arch, or "orbital angle", is a parameter showing distinction between the wolf skull and the dog skull, especially in dogs of similar morphology (e.g., German Shepherd dog). This angle is taken into account for the distinction between the two types of animals; it measures 39–46° in the wolf and 49–55° in dogs [14,15] (Figure 1).

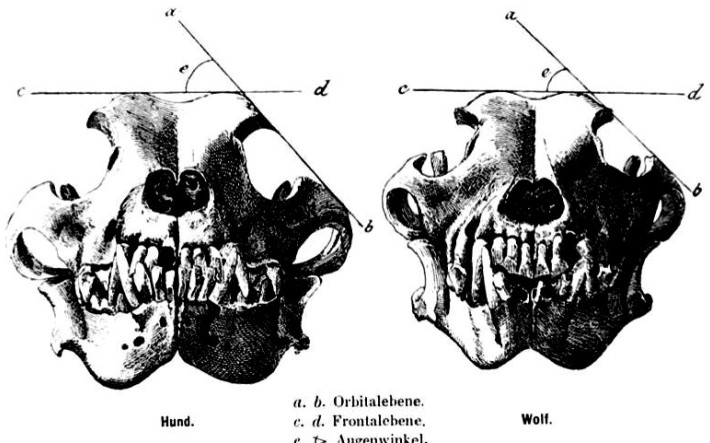

**Figure 1.** Orbital angle as depicted in the original Studer publication (1901).

The orbital angle justifies the different development of chewing musculature in the wolf, which has its own attachment between the considered bones.

The angle between the nasal and frontal bones, or "frontal stop", is a parameter that helps in the identification of the skull specimens; in the wolf, a flatter frontal angle than that in the dog is present (Figure 2).

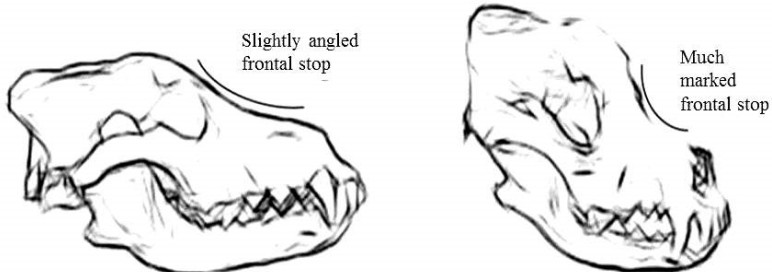

**Figure 2.** Drawing of "frontal stop" of the wolf skull (**left**) and the dog skull (**right**). Copyright © Davide Prinetto.

The wolf adult dental formula is the same as that of the dog, but the canines and "feral" or "carnassial" teeth (PM4/M1) stand out in respect in comparison to those of the dog [16]. An earlier review showed that [17], compared to a wolf of the same size, the dog shows a lighter skull, smaller teeth, wider palate but larger neurocranium. These propositions can be explained by artificial selection.

Currently, there is no motivation related to domestication regarding the shape of the coronoid process of the jaw (on which the temporal and masseter muscles are inserted), which is curved backwards along the ascendant branch in the dog skull (Figure 3).

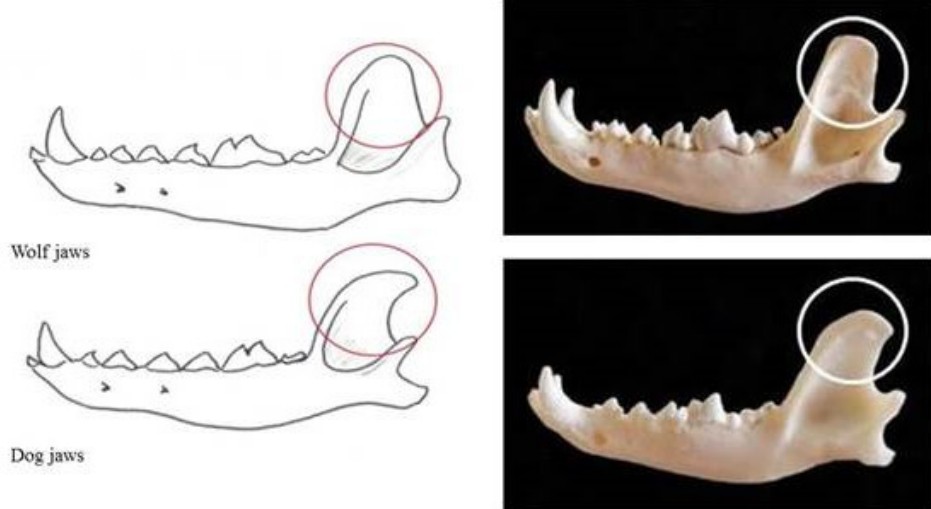

**Figure 3.** The coronoid process of the wolf jaw and the dog jaw. (**left**), Drawing Copyright © Davide Prinetto; (**right**), bone findings (Veterinary Anatomy Museum, University of Pisa).

This study has the purpose of carrying out a morphological investigation of bone specimens of the ancient Italian wolf and dog bone specimens from the Veterinary Anatomy Museum (University of Pisa) to check for significant variables in the two types of bone specimens, as described in the literature.

## 2. Materials and Methods

The authors performed a comparative study on the bone specimens of *Canis lupus* L., chronologically placed in the Wurm period (last glaciation) that were recovered from the archeological dig of Agnano (Pisa) and four osteological skeletal specimens and ten skulls of *Canis lupus familiaris* L. that were kept in the Veterinary Anatomy Museum (University of Pisa), datable around 1850 and of which, in the museum archives, there is no documentation regarding the breed.

The wolf specimen included the skull with the jaw, segments of thoracic and pelvic limbs with a part of the coxal. In the skull, the profile of the nasal bone, the development and profile of the sagittal crest, the measure of the orbital angle, the position of the orbital cavity with respect to the median plane of the skull, the extension of the zygomatic arch, the length of the cranial cavity and the jaw bone processes (jaw body and coronoid process), the width of the cranial cavity (widest interparietal distance) and the depth of the masseteric pit were studied and were compared with the same bone processes of the dog specimens. The morphology of the wolf teeth was compared to that of the dog specimens.

The bone segments relating to the appendicular skeleton of the wolf include the fore-limb bones (humerus, radius and ulna, carpal bones, five metacarpal bones and phalanges of the hand) and the hindlimb bones (tibia, fibula, tarsal bones, five metatarsal bones and phalanges of the foot) and were comparatively evaluated with the same topographical findings of the four dog skeletons kept in the Museum.

### 3. Results

The structure of the wolf skull shows the typical features of a predator; the sagittal crest is very well developed to allow a broad attachment of the temporal muscle, which is more developed in carnivores (Figure 4).

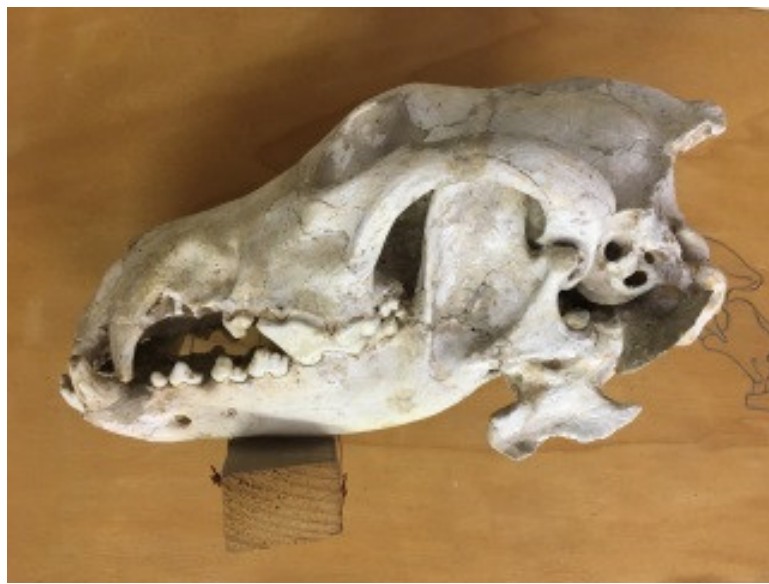

**Figure 4.** The wolf skull (Department of Archeological Sciences, University of Pisa), lateral view.

The dog skulls found housed in the Museum show sagittal crests that are less developed than that of the wolf, and they tend to decline gradually in mesaticephalic breeds and disappear in brachycephalic ones (Figure 5).

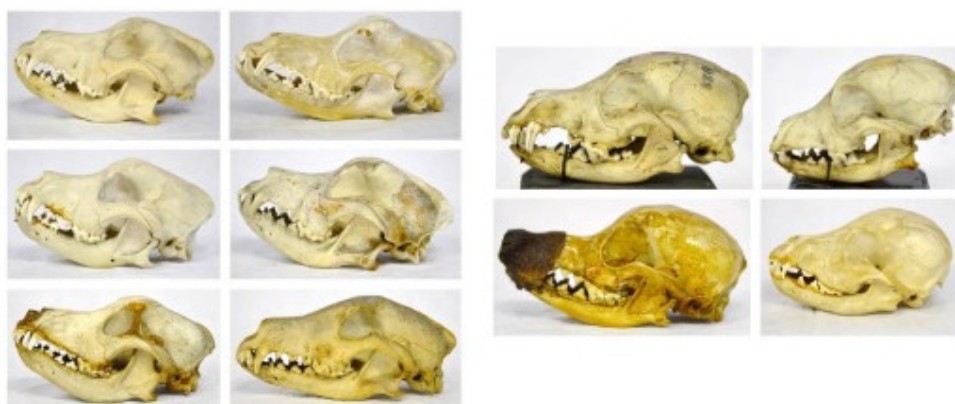

**Figure 5.** Dog skulls (Veterinary Anatomy Museum, University of Pisa).

Due to this, the temporal muscle has a lower efficiency of contraction in some dog breeds due to domestication.

The wolf nasal bone is long and wide and is continuous with the frontal bone, compressed dorso-ventrally (Figure 6).

With the progressive shortening of the splanchnocranium, due to domestication, the dog nasal bone shows an obtuse angle with the frontal bone (called "stop") in dolichocephalic breeds, and it tends to change to an acute angle in mesaticephalic and brachycephalic breeds (Figure 7). The sagittal crest disappears in brachycephalic breeds.

In the wolf skull, the acute orbital angle, between the tangent line to the top of the skull and the zygomatic orbital line, measures 40°. The orbital cavity is more open in the

frontal position because the ventral zygomatic margin of the orbit, which is almost straight, is seen higher than and close to the zygomatic process of the frontal bone (Figure 8).

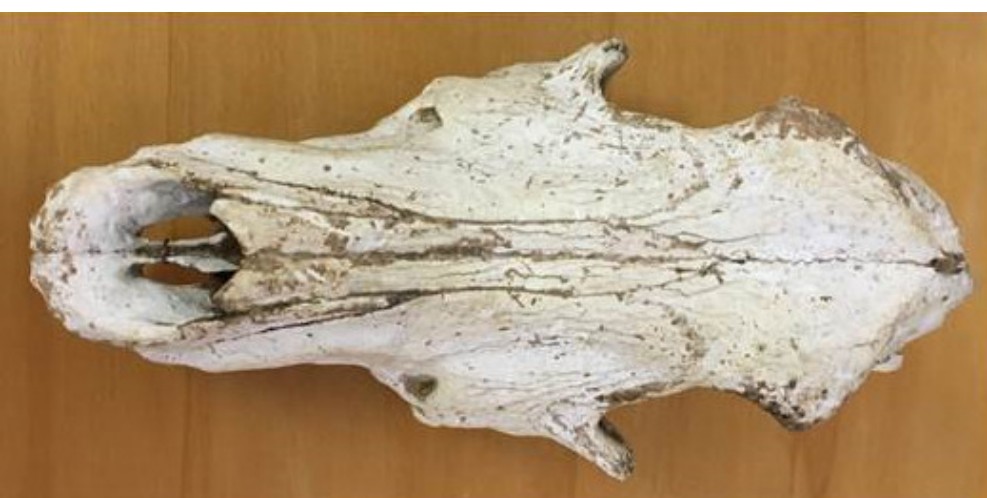

**Figure 6.** The wolf skull (Department of Archeological Sciences, University of Pisa), dorsal view.

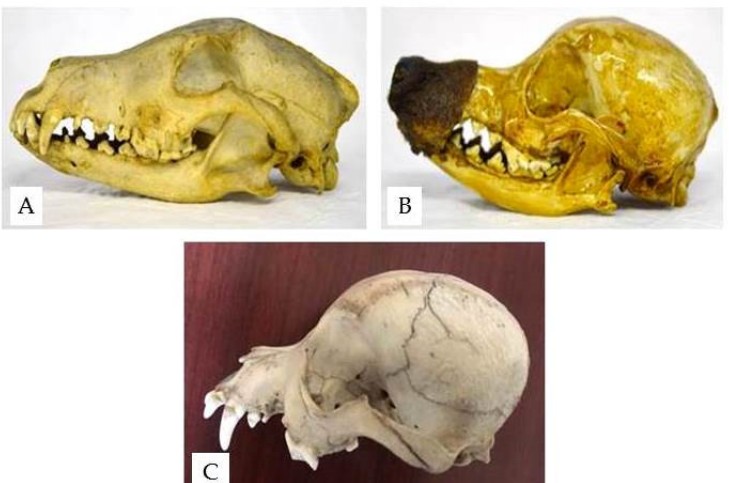

**Figure 7.** Angle between the nasal and frontal bones ("stop") in dolichocephalic (**A**), mesaticephalic (**B**) and brachycephalic breeds (**C**) (Veterinary Anatomy Museum, University of Pisa).

The zygomatic process of the frontal bone expands on the lateral plane. The cranial cavity, measured by a line between the orbital cavity and the occipital bone, is 125 mm in length and 60 mm in width.

In the dog skull, the orbital cavity has a more rounded shape since the ventral zygomatic edge has a more concave profile, allowing a position that is more lateral to the eyeball (field of view less than 180°). The zygomatic process of the frontal bone progressively reduces until it disappears in brachycephalic breeds (Figure 9).

The orbital angle is measured to be between 53° and 60° in the examined bone specimens. The cranial cavity's length and width are 97–99 mm and 53–56 mm for brachycephalic breeds, 101–103 mm and 56–57 mm for mesaticephalic breeds and 123–124 mm and 57–62 mm for dolichocephalic breeds, respectively.

In the wolf jaw, the masseteric pit (point of insertion of the masseteric muscle) is deep and the profile of the jaw body is linear with the coronoid process (point of insertion of the temporalis muscle); it is broad and rounded at its apex, which diverges laterally, and is in connection with the breadth of the zygomatic arch. Due to shortening of the splanchnocranium during domestication, the profile of the dog jaw body becomes progressively more convex, with the maximum degree present in brachycephalic breeds (Figure 10).

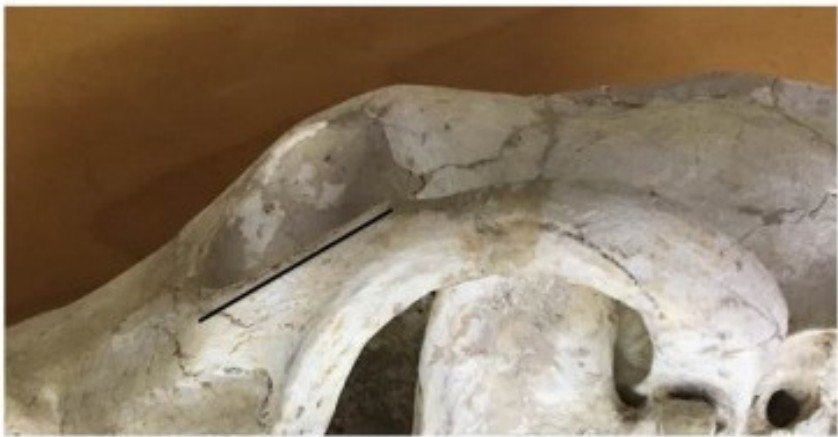

**Figure 8.** Ventral zygomatic margin of the wolf orbit (inserted line) (Department of Archeological Sciences, University of Pisa).

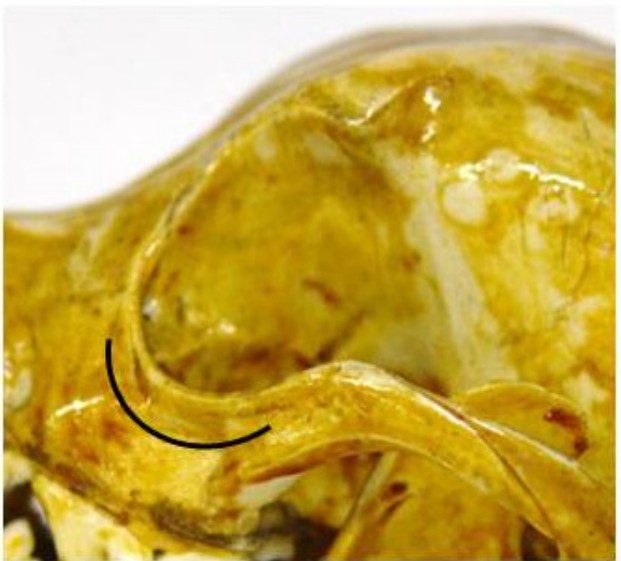

**Figure 9.** Ventral zygomatic margin of the dog orbit (inserted line) (Veterinary Anatomy Museum, University of Pisa).

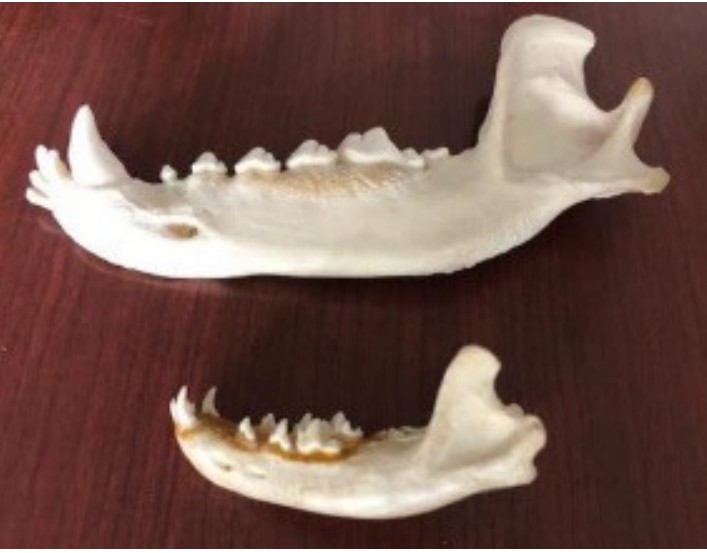

**Figure 10.** Dog jaws in dolichocephalic (**up**) and brachycephalic (**down**) skulls (Veterinary Anatomy Museum, University of Pisa).

The masseteric pit is shallower than that in the wolf, and the coronoid process is more slender and caudally curved in the profile of the jaw body.

The wolf teeth are typical of a carnivore, with very well developed canine teeth and "feral" or "carnassial" teeth, suitable for slicing and keeping the prey firmly in the mouth. The process of domestication has not resulted in profound changes in the dog teeth; they are typical of a carnivore, with a lateral overlap of the maxillary teeth over the mandibular ones during occlusion.

A deviation in the placement of the teeth results as the sizes of the teeth do not decrease proportionately with a reduction in the length of the jaw; as a result of the shortening of the two bone arches in brachycephalic breeds, the teeth appear closer together, reducing the sizes of the canines and feral teeth, due to a different food supply compared to the wild progenitor [18] (Figure 11).

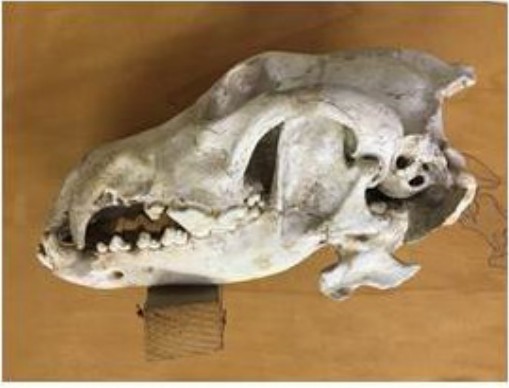
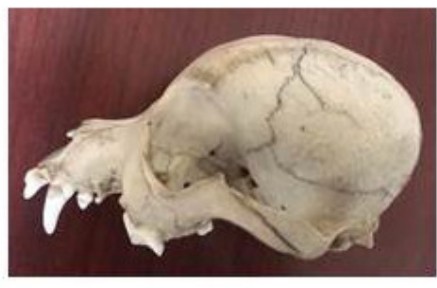

**Figure 11.** Wolf teeth (**left**) (Department of Archeological Sciences, University of Pisa) and brachycephalic dog teeth (**right**) (Veterinary Anatomy Museum, University of Pisa).

The appendicular skeleton, which includes the wolf forelimb and hindlimb, shows bone segments similar to those in dog skeletons of comparable size; the epiphyses and diaphyses of long bones and the hand and foot bones have the same anatomical features for muscle and ligament attachment and for the joint surfaces for diarthrosis (Figures 12 and 13).

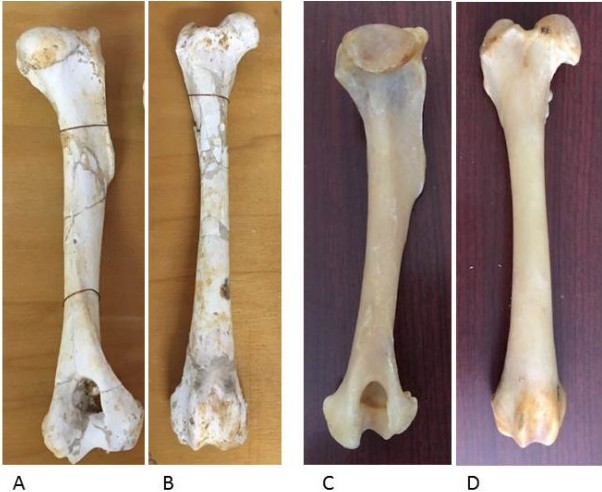

**Figure 12.** Wolf humerus (**A**) and femur (**B**) (Department of Archeological Sciences, University of Pisa). Dog humerus (**C**) and femur (**D**) (Veterinary Anatomy Museum, University of Pisa).

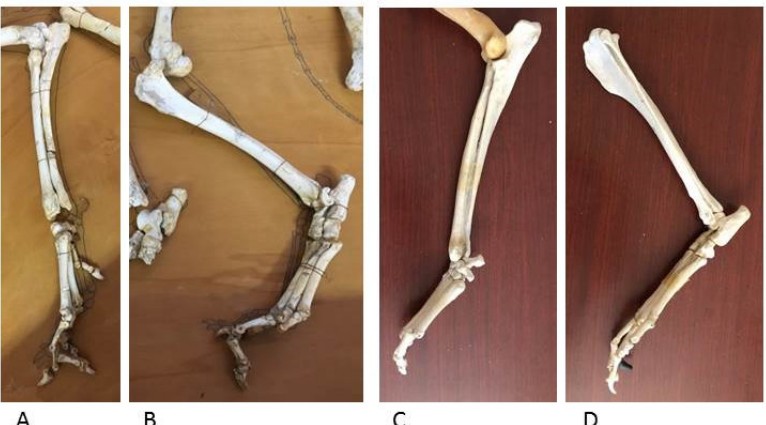

**Figure 13.** Wolf distal forelimb (**A**) and hindlimb (**B**) (Department of Archeological Sciences, University of Pisa). Dog distal forelimb (**C**) and hindlimb (**D**) (Veterinary Anatomy Museum, University of Pisa).

## 4. Discussion

The transition from wolf to dog is observed by the study of skeletal features, accompanied by a series of parameters useful in understanding this biological passage. A diminution in size occurs early in the process of animal domestication; this phenomenon is characterized in many animal species, not just the dog.

This work suggests that, in order to differentiate ancient wolf skeletal specimens of *Canis lupus* L. and similar specimens of *Canis familiaris* L., it is necessary to carry out a series of morphological investigations, since there is not one single significant parameter that can be used alone for this investigation.

In the skeletal study, the skull features of the splanchnocranium and neurocranium bones are more interesting than the skeletal features in relation to the appendicular skeleton detected in the lupine subject of the archaeological excavation. In the wolf skull, almost all of the studied anatomical parameters show different features with respect to the same features in the dog specimen. In particular, the morphology of the sagittal crest and the nasal bone, the position of the orbital cavity and the measurement of the orbital angle and the morphology of the masseteric pit proved to be valuable features in differentiating the wolf skull from the dog skull.

The similar features in the wolf and dog teeth indicate that the domestication process did not change the approach to food, which remains typical of a carnivorous animal even after the transformation of the dog into an omnivorous animal. The teeth are therefore more conservative and remain large. Crowding of the teeth and overlapping of cheek teeth is diagnostic of the early domesticated dog compared to wolves [18].

Different developments of the previous parameters consequently indicate a modification in the development of the neurocranium. The analysis of cranial length and width, in the different types of breeds of the analyzed dog skulls, is a useful parameter for discriminating the wolf and dog skulls; indeed it is pointed out that in a dog that is also dolicocephalic, the neurocranium is generally wider and shorter in size, in accordance with some studies in the literature [19] that refer to this feature as being related to the reduction in the development of the limbic system and rhinencephalon during domestication. The morphological differences highlighted could also be attributed to the phenomenon of neotenic pedomorphism, i.e., the conservation in adult dogs of morphological and behavioral traits typical only of different juvenile stages of wolf development, as a result of the selection processes following the domestication process.

Instead, the anatomy of the appendicular skeleton does not vary in the organization of the long or short bones that characterize it; beyond the variable length and morphology of the diaphysis in different dog breeds, domestication has not led to structural variations.

The lack of evidence of remarkable changes in skeletal anatomy from Pleistocene (the geological time to which the examined wolf skeleton belongs) to Upper Paleolithic suggests that no dependent relationship has yet been established between wolves and humans. Casual association must have occurred if a wolf puppy would sometimes be kept by a human family [18].

The only skull differences found might derive from the different types of lives of wolves that would have approached humans; an animal integrated into the world of humans does not need to kill for autonomous survival, leading to a reduced brain capacity and therefore to a change in the morphology of the skull. Since these conclusions are valid only for the small group of subjects included in this work, further studies are needed to validate these conclusions.

**Author Contributions:** Conceptualization, A.C. and E.G.; methodology, A.C.; investigation, D.P.; resources, A.C.; writing–original draft preparation, A.C.; writing–review and editing, A.C.; supervision, E.G. All authors have read and agreed to the published version of the manuscript.

**Funding:** This research received no external funding.

**Institutional Review Board Statement:** Ethical review and approval were waived for this study due to the fact that for the anatomical observations in a wolf skeleton (from archeological site) compared with dog skeletons (preserved in the Veterinary Anatomy Museum of the University of Pisa), no animal manipulation was carried out on living subjects. The study does not fall within the typology for which it is necessary to comply with the indications of the European legislation that regulates its use (Directive 2010/63/EU on the protection of animals used for scientific purposes).

**Informed Consent Statement:** Not applicable.

**Data Availability Statement:** Not applicable.

**Acknowledgments:** All individuals included in this section have consented to the acknowledgement of Department of Archaeological Sciences, University of Pisa.

**Conflicts of Interest:** The authors declare no conflict of interest.

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
