# Peer review of "Wolf and Dog: What Differences Exist?"

_2813-0545, doi:10.3390/anatomia2010007_

Round 1
Reviewer 1 Report
The manuscript is enough well written, anyway some changes have to be performed in the structure before going ahead in the pubblication phase. If authors well follow the suggestion given below I will certainly recommend this case report for pubblication on the Journal.
Firstly, I suggest you to imporve the introduction section because in its current form is a little bit too short. In particular, I suggest you to add some informations about genetic differences between dog and wolf. I known that is not the aim of the work but this aspect is reported in the article and for this reason it's important to describe it better (just few lines). In order to realize this I suggest you to include these works
-https://doi.org/10.1111/j.1365-294X.2007.03262.x
- https://doi.org/10.1016/j.biocon.2020.108525
Moreover, I strongly advice you to include in the introduction section that the similarity between dog and wolf have an important role in disease's spread and how most of pathogens that afflict wolfs can pass to dogs and viceversa. Concerne this aspect, it's actually to show the potentiality of GIS in study of zoonoses. Therfore I suggest you to include these works:
- https://doi.org/10.3390/rs12213542
- https://doi.org/10.3390/ani12081049
- https://doi.org/10.1186/1751-0147-42-S1-S79
Reviewer 2 Report
This paper is interesting and a pleasure to read. However, I suggest:
- -use the term “it was done” instead of “we did”,
- keywords should not be a duplication of words from the title,
- M&M must accurately identify the material being tested. How many individuals of dogs were analyzed?, what do we know about them (at least the breed)? Please expand on the description of the measurements taken and the features analyzed.
- Please consider a cranial cavity analysis. We know that with domestication there are some changes in the brain including the limbic system. A metric analysis of the cranial cavity would be valuable (e.g., using CT).
- Did the number of teeth follow the dental pattern in all brachycephalic skulls? Especially in the mandible, a reduced number of teeth is sometimes observed.
- The analysis of the skull is so interesting and extensive that I propose to remove the section on other bones, especially since it is not described in sufficient detail. It should also be discussed more extensively.
- citations should be numeric (line 207),
- discussion needs improvement. It's hard to call it a discussion if there is one quote in it.
Round 2
Reviewer 2 Report
It looks much better. I further believe that the discussion should include more references, but the paper is valuable and should be published in a journal.